# Effects of Vegetation Structure on Psychological Restoration in an Urban Rooftop Space

**DOI:** 10.3390/ijerph20010260

**Published:** 2022-12-24

**Authors:** Juyoung Lee, Minji Kang, Sungku Lee, Seoyong Lee

**Affiliations:** 1Department of Landscape Architecture, Hankyong National University, 327 Chungang-ro, Anseong 17579, Republic of Korea; 2Graduate School of Environmental Studies, Seoul National University, 1, Gwanak-ro, Gwanak-gu, Seoul 08826, Republic of Korea

**Keywords:** urbanization, green space, stress reduction, heart rate variability

## Abstract

Connectedness to nature has been recognized as an important factor for well-being, with rooftop green spaces being used for stress reduction in modern cities. This study aimed to examine psychological and physiological responses to three different vegetation models on an urban rooftop. An analysis of psychological parameters indicated that the existence of vegetation in rooftop spaces could have positive effects on mood states, and the size of the effect was greater in a structured vegetation design than in a monotonous one. An analysis of the physiological parameters of heart rate variability and systolic blood pressure indicated that greater restorative effects are elicited from the use of vertical elements, such as shrubs and trees, added to grassy areas than from concrete environments. However, a fully enclosed vegetation with trees was not associated with higher parasympathetic activities than a half-open vegetation model. Based on these findings, an open and structured vegetation design that includes both grass and shrubs may have more potential for stress reduction than a monotonous vegetation model. A larger volume of vegetation was not necessarily linked to higher psychological and physiological benefits.

## 1. Introduction

In the past several decades, notable landscape changes have been seen around the world as a result of increasing urbanization. The global urbanization rate rose from 30% in 1950 to 56% in 2018, and the share of urban populations is continuously increasing in all regions around the world [1]. A higher rate of urbanization is associated with less opportunity for contact with the natural environment in daily life. In addition, the prevalence of physical inactivity and a sedentary lifestyle reduces the likelihood of participating in nature-centered activities. 

Stress reduction theory [2] and attention restoration theory [3] illustrate how nature can influence an individual’s psychological response to a stressor. Connectedness to nature has been recognized as an important factor in well-being [4], as seen in reports investigating the relationships between urbanization and health issues [5,6,7,8]. A substantial body of research supporting the health benefits of natural settings has attracted interest in the development of nature-friendly urban environments [9,10,11,12]. According to population studies, exposure to natural environments is linked to reduced health inequality [5] and to increased longevity of urban seniors [13]. Green space consisting of natural elements also positively influences youth development [14]. Green space vegetation can promote cardiovascular health and reduce overweight by encouraging physical activity [15]. With growing evidence of the health benefits of green space [16,17,18], questions regarding the practical use of green space in urban area have been raised. Many cities, in fact, are currently looking for nature-based solutions to promote public health [19]. A conceptual framework linking urban green spaces with human health has been proposed [20,21], and practical approaches from the perspective of urban planning and design have been suggested [22,23]. 

The concept of urban green spaces has been adopted in the design of modern buildings in many ways, with the intent of increasing the environmental, ecological, social, and commercial functions of the building. A building’s rooftop is an ideal location for a green space with open views, providing people with opportunities for relaxation, recreation, and social interaction. Rooftop green spaces are also used therapeutically in hospital and welfare facilities due to their intrinsic capacity for stress reduction and restoration [24,25]. The health effects of green space are related to the type of green space which can be characterized by vegetation features [26]. An epidemiological study investigating psychological distress levels when exposed to different types of vegetation indicated that total green space and percent cover of tree canopy could be important factors in reducing psychological distress [27]. An experimental study compared the psychological effects of three different vegetation types—wooded parkland, tended woodland, and wild woods—by evaluating self-reported restoration levels [28]. The results indicated that different degrees of naturalness in wooded landscapes could not necessarily be linked to the different potentials for psychological restoration. A Chinese study using virtual reality images examined the restorative potential of different environmental components, including concrete, grass, and trees [29]. The findings showed that an environment with short-grass ground cover had stronger restorative effects than one with trees or no vegetation. A physiological study measured changes in cardiac output in American older adults when they were exposed to differently structured vegetation types, namely a herb garden, a structured Japanese-style garden, and a simple landscaped area planted with a single tree [30]. The results showed that the green space with structured vegetation evoked greater responses in the outcome measures than the green spaces with single-layered vegetation or a single tree. Lindal and Hartig (2015) examined the relationships between multiple types of natural elements and the likelihood of physiopsychological restoration in an urban streetscape [31]. Trees, grass, and flowers were selected as vegetation variance for restorative design in computer simulation images, and streetscapes with more trees were found to be likelier to have higher restoration values. Based on previous studies, the potential for stress reduction and restoration in green space seemingly relates to the structural characteristics of the vegetation. A previous study [32] investigated the association between stress response and vegetation density in woodland, demonstrating that a higher density of vegetation could cause more positive psychological outcomes. However, few studies have investigated the associations between stress reduction and vegetation characteristics.

In the field of green space planning, an evidence-based approach has been recommended to develop vegetation models for healthier green spaces [33]. Robust evidence is required to have a better understanding of the associations between green space characteristics and psychological outcomes, which may reduce the gap between research and design practice [34]. Recent empirical studies have posed meaningful challenges in dealing with this issue. Nordh et al. (2009) suggested vegetation design guidance for small urban parks by investigating the association between the physical components of a park and its restorative capability [35]. Hadavi et al. (2015) took a more practical approach to green space design in urban residential areas [36]. They investigated preferences for 93 neighborhood photos and illustrated that garden spaces filled with vegetation were the most preferred scenes. The results of these studies indicate that vegetation characteristics could be a critical factor affecting the restorative capability of small green spaces. The vegetation characteristics of urban green spaces are largely affected by planting design, because the selection of plant assemblages determines species composition and vegetation structure [37]. In particular, a vertical structure is closely associated with visual properties and the volume of green, which is determined by vertically layered vegetation strata. Insufficient data are available on the relationships between vegetation structure and restoration capability.

The development of computer graphic technologies enables effective testing of the validity of green space planning. In landscape design, simulation techniques have been applied to design practice because high-resolution computer images can present highly realistic virtual environments [38]. Moreover, in empirical studies, simulated computer images or virtual environments have been utilized to measure emotional responses to diverse types of natural environments [39,40,41,42]. Despite the gap between a real natural environment and a simulated natural environment [43], simulation methodology has the advantage of evaluating urban green space more precisely and assessing user preferences for vegetation designs.

Despite a large body of research, an empirical approach bridging scientific knowledge and design practice is still lacking. Insufficient evidence is available on how the design of vegetation in a specific green space can be used to increase restoration potential. Therefore, we examined the effects of vegetation structure in an urban rooftop space on restorative outcomes. Because vegetation structure influences the landscape and scenic characteristics in a rooftop space, three differently layered vegetation models were considered in this study. To evaluate the psychological and physiological benefits of rooftop green space models, we used a computer simulation technique that has been verified in previous research [38,39,40,41,42,43].

## 2. Methodology

### 2.1. Visual Stimuli and the Research Process

The effects of different rooftop models on psychological restoration were studied through a within-group analysis. Landscape simulation techniques were adopted to create realistic visual images, which have been verified through many previous studies [31,39,41]. The original photographic image was taken of a typical rooftop space of a 15-floor commercial building in city center of Seoul, Korea. The rooftop had enough space for vegetation (1860 m^2^) and a concrete floor. All the rooftop images had the same background of buildings so as not to provoke aesthetic emotions because built environment types can also influence stress levels [29]. Using the original image with no vegetation (Control), three different vegetation models were simulated using computer software (Adobe Photoshop CC 2018, version 19.0). Green rooftop images with different vegetation structures were prepared on the basis of planting design principles [37], including single-layered, two-layered, and three-layered planting models (Table 1). Vegetation structure is also related to planting types of grass, shrubs, and trees that are critical elements necessary for a restorative experience [28]. In addition to the control, the other three images were as follows: single-layered Grass type (G), in which there was mainly ground cover vegetation and a deck for walking; two-layered Grass+Shrub type (GS), in which shrubs were planted in addition to grass vegetation; and three-layered Grass + Shrub + Tree type (GST), in which tall trees were added to the GS type (Figure 1). Each landscape image had a different volume of green: 0% in the control, 30% in G, 50% in GS, and 70% in GST. No benches or pergolas were added to the simulation images to exclude artificial factors that might affect the participants’ visual perceptions. All the plant foliage was green, and no flowers were added to avoid introducing the effects of color.

An experiment using computer-simulated images was performed in a laboratory with a temperature regulation system. Before the experiment, the study objectives and data collection procedures were explained to all the participants. At the beginning, an electrocardiogram (ECG) sensor (My beat, Union tool, Tokyo, Japan) was attached to the left side of the participant’s chest to collect heart rate variability (HRV) data as they viewed each image. To investigate the physiological and psychological responses to different vegetation models for the rooftop green space, the participants were shown four different images. Each image was presented for 90 s using a high-resolution monitor. The images were presented randomly to reduce order effects, and 30 s with no image was included between the images. Prior to the experiment, the participants were shown a dummy image to reduce psychological tension at the beginning of the session. HRV was monitored continuously throughout the experiment. Systolic and diastolic blood pressure was measured using a blood pressure monitor (HEM-1000, OMRON, Kyoto, Japan) after viewing the images. Psychological questionnaires were administered after the participants’ blood pressure was measured. Semantic differential (SD) methods and a Korean version of the Profile of Mood States (K-POMS) were used to investigate perceptions and mood changes after viewing the four different rooftop images.

### 2.2. Subjects

Fifteen male adults with a mean age of 23.7 ± 1.7 (standard deviation) years participated voluntarily in this study. All subjects were confirmed to be free of diagnosed allergies, cardiovascular or mental disease, and alcoholism. Informed consent was obtained from all subjects prior to beginning the study. Ethical approval for this study was obtained from the Public Institutional Bioethics Committee (P01-202009-12-002).

### 2.3. Data Collection and Analysis

As a parameter for psychological state, the SD consisted of four items—natural–artificial, open–closed, attractive–unattractive, and comfortable–uncomfortable—investigated by statements rated on a seven-point scale. The K-POMS included question items examining changes in mood states, such as tension–anxiety, anger–hostility, depression, vigor, fatigue, and confusion. In the analysis of HRV data, parasympathetic and sympathetic nervous activities were calculated based on the values of high frequency (HF; 0.15–0.4 Hz) and the ratio of low frequency (LF, 0.04–0.15)/high frequency (LF/HF), respectively [44]. Analysis of variance (ANOVA) and post hoc Scheffé multiple comparisons were conducted to identify statistical differences among the psychological and physiological responses to the four images. Statistical analysis was performed using SPSS 21.0 (IBM-SPSS Statistics, Armonk, NY, USA), and the significance level was *p* < 0.05.

## 3. Results

Significantly different values were detected in some physiological parameters among the responses to the landscape images (Table 2). In the analysis of HRV data, a significant difference was found among responses to the different images. The type GS (615.5 ± 63.0) had a significantly higher HF value of HRV than the control (327.4 ± 60.1; *p* < 0.05), with no significant differences among other types (Figure 2). There was no significant difference between the control and the GST design, which had the highest volume of green vegetation. In the analysis of the LF/HF ratio of HRV, no significant differences were found among the four images. In the analysis of blood pressure as a parameter of autonomic nervous activity, different patterns of value changes in response to the four images were also found. Significantly lower values of systolic blood pressure were detected in response to the type GS (119.5 ± 1.9) and type GST (121.1 ± 2.2) images compared to the control (134.0 ± 2.9; *p* < 0.01; Figure 3). No significant differences in diastolic blood pressure or pulse rate were identified.

An analysis of the psychological responses to the four images using SD methods showed some significant differences in the visual perception of the vegetation designs. Significantly higher scores for naturalness, attractiveness, and comfort were found in response to the GS and GST images compared to the control (Figure 4). Despite a higher mean value of type G than the control, no significant difference was detected in any SD subscale. An analysis of the POMS data supported the premise that rooftop vegetation could change viewers’ mood states in a positive way and that mood state could differ according to the vegetation design (Figure 5). The scores of negative subscales, such as tension-anxiety, anger-hostility, depression, fatigue, and confusion, decreased significantly more in the G, GS, and GST images than in the control. In addition, the level of vigor increased more after viewing the non-control images compared to the control image. Types GS and GST showed significantly lower values in the subscales of anger-hostility and fatigue and higher scores in vigor compared to type G. Significant differences among vegetation types in POMS subscales were also detected, with lower scores in type GS than in type G in the negative mood states of anger, fatigue, and confusion.

## 4. Discussion

In this study, psychological and physiological responses to different vegetation structures in a rooftop green space were examined to find evidence for an optimal green space model for stress reduction. An analysis of psychological parameters indicated that green spaces change mood states in a positive way by decreasing negative feelings and increasing positive ones, which is consistent with previous findings looking at woodland areas [45,46,47]. A grass-only type with the most monotonous vegetation structure significantly increased vigor score when compared to the image of a concrete rooftop. A previous simulation study had similar results, in which grass could be seen to facilitate restoration by having increased positive effects on stress [29]. In the present study, the different landscape images produced different vigor levels, which partly concurs with previous findings that different natural environments have different capacities to affect positive emotion [48,49]. Relatively higher positive emotion scores were found for the structured vegetation models than for the grass model. Responses in terms of negative emotions also varied depending on the vegetation model. Significantly lower levels of anger, fatigue, and confusion were found in response to the types GS and GST, which had two- or three-layered vegetation structures, than the type G with ground cover only. Thus, our results indicate that vegetation in rooftop influences affective responses, and that vertically structured vegetation models are likely to be associated with greater psychological benefits.

The results of analyses of the physiological parameters of HRV and systolic blood pressure indicated that greater restorative effects were elicited when the participants were exposed to a structured vegetation environment, such as that of types GS and GST, than to a concrete environment. The physiological outcomes in this study partly align with previous studies investigating autonomic nervous activity when exposed to natural elements [9,50]. This result cannot be explained by the nature versus urban dichotomy, because physiological benefits were detected only in vegetation structures that include shrubs. These findings could be considered from the perspective of green volume, which refers to the amount of green in the leaves, branches, and stems compared to the entire landscape [51]. The amount of greenery in a view can influence affective states and physiological outcomes [52]. A previous study showed positive correlations between the amount of greenery and alpha wave activity in the brain by investigating EEGs [53]. However, some studies showed different results that increasing amount of greenery is not always linked to the increasing value of subjective preferences and psychological effects in indoor [54] and outdoor [55] spaces. Given that the highest HF value was detected in the type GS, with 50% green volume, rather than in the type GST, with 70% green volume, it is clear that restorative potential is not aligned with increasing amounts of vegetation.

An explanation for the more positive effect of the type GS might be its structured vegetation and more open landscape. Goto et al. (2013) insisted that a structured green space had more capacity for having physiological effects on the elderly population than a green space composed of grass or a single tree [30]. Previous research has demonstrated that bushes or shrubbery could be important elements for restoration in green spaces [28]. In the results of the current study, psychological and physiological parameters changed in a positive way when shrubs were added to grass vegetation. Thus, the structure of vegetation may affect the restorative properties of green spaces, and shrubs may increase the likelihood of restoration by improving vegetation structure. Trees also increase the density of vegetation and add to the vertical structure of green spaces. Nevertheless, dense tree vegetation is unlikely to improve psychological effects because there was no evidence that the type GST had a significantly more positive effect on psychological and physiological parameters in our study. Similarly, Chiang et al. (2017) reported that subjective preference and psychological effects did not increase with greater density of trees [32]. Tree density is related to the openness or preference of a landscape. Sometimes, dense vegetation of trees might act as a problematic visual barrier by blocking visual expansion [56]. Recent research has indicated that fully enclosed green spaces are perceived as less restorative than open spaces [28,57,58]. Based on this finding, open and structured vegetation model, such as the type GS, may have more potential for stress reduction in an urban rooftop space than a monotonous or fully enclosed vegetation models.

This study focused on exploring the effects of different vegetation structures on psychological and physiological responses in a small urban space. The planting design of green space is a complex process because many elements, such as species, shapes, colors, sizes, and allocation patterns, should be considered in practical work, as indicated in previous studies [29,31,36,37,59]. Finding a preferred model of vegetation structure could make it possible to anticipate the outline of planting density and species selections for restorative green space, because the vegetation structure reflects the output of these factors. The findings from this pilot study, however, might not be applicable to street areas. As the preferred vegetation model can differ depending on the spatial size or shape in cities [31] we need to find additional evidence for planning and design solutions in practice.

There were some limitations to this study. To reduce the variables that stem from different urban backgrounds, images were created based on one sample site. Thus, the results of this study cannot be extended to other scales and types of city landscapes. In addition, the findings cannot be generalized because the sample size was relatively small and limited to young adult males. The experiment was carried out in the laboratory and the experiment design was kept simple to control variables that might affect physiopsychological outcomes. To increase the validity and practical applications of our results, further studies should consider expanded sample groups and diverse compositions in vegetation design, including flowers [31] and other plant species [60].

## 5. Conclusions

A controlled laboratory experiment was conducted to compare the psychological and physiological effects of three different vegetation models for a rooftop green space. Relatively precise vegetation designs were prepared using graphic techniques to evaluate the possibility of stress reduction of practical green space design. This was a pilot approach for considering the connection of science and design work for making restorative green space. Our data showed that vegetation structure could influence the quality of restoration in green spaces. The vertically layered vegetation model had higher potential for stress reduction and restoration than a monotonous vegetation model for an urban rooftop. Our results partly align with previous theories and studies finding that vegetation can have positive effects on psychological restoration in urban areas [2,3,17,45]. However, the fully enclosed vegetation model did not support a greater stress reduction effect. With increasing interest in evidence-based green space design, studies have been conducted to bridge research findings and design practice [35,61]. To provide more useful information for design practice, more precise guidance or outlines need to be suggested. In this context, our findings contribute to a better understanding of the restorative vegetation model. However, the findings from our pilot approach are not generalizable to extended spaces in cities. The preferred vegetation model can differ depending on the shape of the site, such as an open square or linear street. For example, a large number of trees were preferred in a streetscape [31], but not in our test site. Although the complexity of the design process makes it difficult to apply our knowledge to drawing work, empirical studies on the relationship between restorative potential and various design models will provide us with a clearer vision for using green space for public health.

## Figures and Tables

**Figure 1 ijerph-20-00260-f001:**
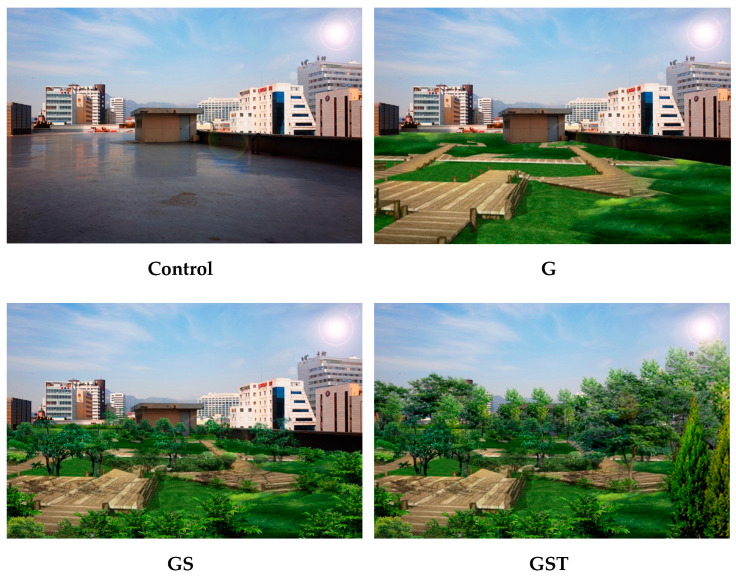
The four types of rooftop landscapes used in this study. Control: no vegetation with concrete floor; type G: grass vegetation; type GS: grass-shrubs vegetation; type GST: grass-shrubs-trees vegetation.

**Figure 2 ijerph-20-00260-f002:**
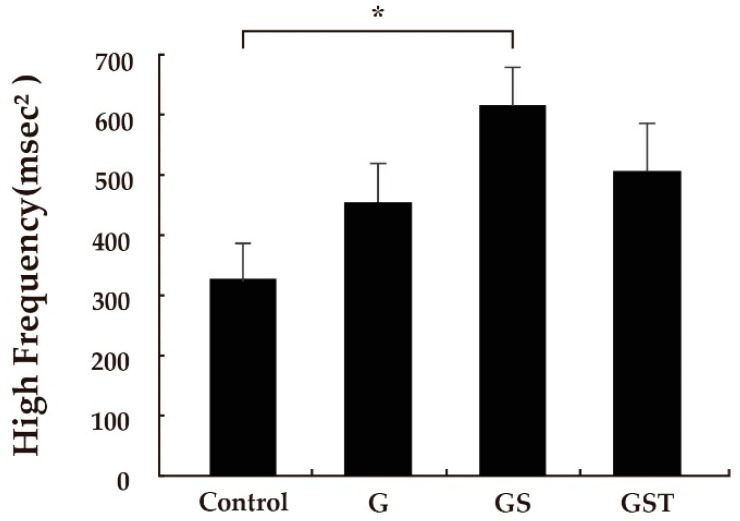
Values of high frequency power when viewing the four rooftop images. Control: no vegetation with concrete floor; type G: grass vegetation; type GS: grass-shrubs vegetation; type GST: grass-shrubs-trees vegetation; *n* = 15, mean ± SE; *, *p* < 0.05.

**Figure 3 ijerph-20-00260-f003:**
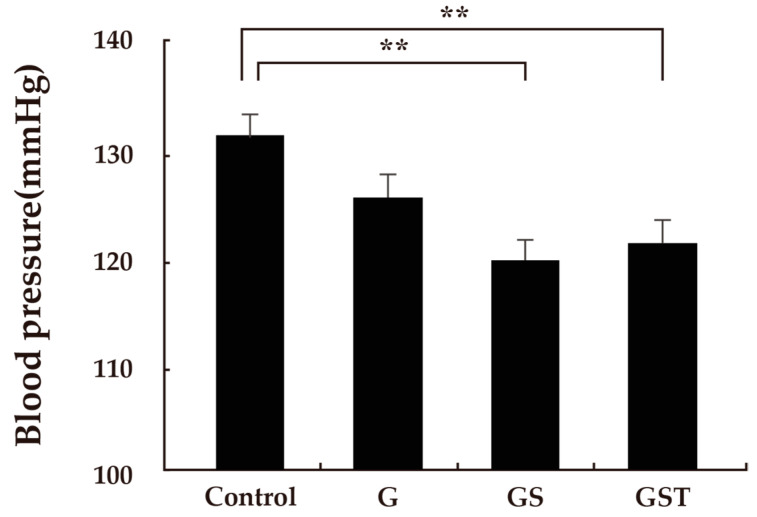
Values of systolic blood pressure after viewing the four rooftop images. Control: no vegetation with concrete floor; type G: grass vegetation; type GS: grass-shrubs vegetation; type GST: grass-shrubs-trees vegetation; *n* = 15, mean ± SE; **, *p* < 0.01.

**Figure 4 ijerph-20-00260-f004:**
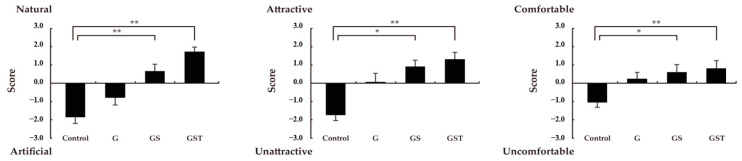
Differences in semantic differential scores among the four rooftop images in terms of feelings of naturalness, attractiveness, and comfort. Control: no vegetation with concrete floor; type G: grass vegetation; type GS: grass-shrubs vegetation; type GST: grass-shrubs-trees vegetation; *n* = 15, mean ± SE; *, *p* < 0.05; **, *p* < 0.01.

**Figure 5 ijerph-20-00260-f005:**
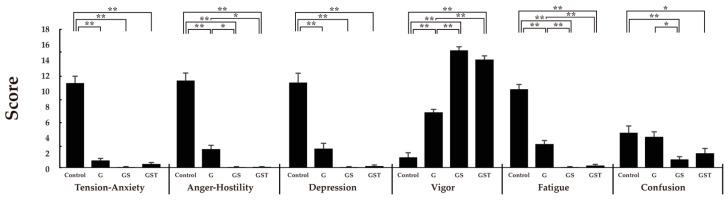
Comparison of Profile of Mood States (POMS) subscale scores for the four rooftop images. Control: no vegetation with concrete floor; type G: grass vegetation; type GS: grass-shrubs vegetation; type GST: grass-shrubs-trees vegetation; *n* = 15, mean ± SE; *, *p* < 0.05; **, *p* < 0.01.

**Table 1 ijerph-20-00260-t001:** Planted species in three vegetation layers; G: grass layer; S: shrub layer; T: tree layer.

Layer	Species	Classification	Height (m)	Plant Form
G	*Zoysia japonica*	Grass or Herb	0.1–0.2	Grass
S	*Euonymus japonicus*	Shrub	1.0–1.5	Round or vase
	*Spiraea* sp.	Shrub	0.5–1.0	Irregular or Weeping
	*Euonymus alatus*	Shrub	1.0–1.5	Vase or Open
	*Rhododendron* sp.	Shrub	0.5–1.0	Vase or Round
	*Buxus mycrophylla var. koreana*	Shrub	0.5–0.8	Round or Vase
	*Sorbaria sorbifolia var. stellipila*	Shrub	0.8–1.8	Irregular
T	*Cornus officinalis*	Tree	2.0–2.5	Vase or Irregular
	*Crataegus pinnatifida*	Tree	2.5–3.5	Oval or Open
	*Sorbus alnifolia*	Tree	2.0–3.0	Round or Oval
	*Taxus cuspidate*	Tree	1.5–2.0	Pyramidal
	*Thuja orinetalis*	Tree	1.5–2.0	Columnar
	*Cornus kousa*	Tree	2.5–3.5	Open or Oval

**Table 2 ijerph-20-00260-t002:** Physiological values produced by the four rooftop images. Control: no vegetation with concrete floor; type G: grass vegetation; type GS: grass-shrubs vegetation; type GST: grass-shrubs-trees vegetation; *n* = 15, significance verified by ANOVA.

Parameters	Types	Mean	SE	Significance
HRV_HF (ms2)	Control	327.4	60.1	*p* < 0.05
	G	454.2	64.7	
	GS	615.5	63.0	
	GST	506.6	79.1	
HRV_LF/HF (ratio)	Control	4.7	1.2	n.s.
	G	3.0	0.4	
	GS	2.9	0.8	
	GST	2.4	0.4	
Diastolic Blood Pressure (mmHg)	Control	74.0	5.3	n.s.
	G	68.3	2.4	
	GS	66.9	2.0	
	GST	65.6	2.8	
Systolic Blood Pressure (mmHg)	Control	131.1	2.0	*p* < 0.01
	G	125.3	2.1	
	GS	119.5	1.9	
	GST	121.1	2.2	
Pulse rate (beats/min)	Control	77.8	2.6	n.s.
	G	74.4	2.3	
	GS	75.5	2.5	
	GST	75.3	1.2	

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
