# Peer review of "Effects of Vegetation Structure on Psychological Restoration in an Urban Rooftop Space"

_ijerph, 2022, doi:10.3390/ijerph20010260_

Round 1

Reviewer 1 Report

Manuscript “Empirical study of vegetation design types promoting physiopsychological restoration” aims to identify evidence based designs for restorative rooftop green spaces by investigating physiopsychological responses to three vegetation design types.

The manuscript is the methodical study. I recommend publishing it after recommended corrections.

Comments for authors

Language

There are no major remarks to the language. Some sentences are too long.

Subject

Correct topic of the manuscript, consistent with the content.

Keywords

Selected correctly. Can only be completed with the name of the main method/methods.

Abstract

The summary reflects the content of the article. Contains the aim of the study, description of methods and the main results.

 1. Introduction

The introduction is not well established in the world literature. The review is short and omits many methodological studies. I recommend completing this section. There is no general view of the spatial distribution, influence and quality of residential green space.

Moreover, verses 28-34 are a description of conditions in Korea. It is a description of the background (area) of the research. This passage should not be included in the introduction.

 2. Methodology

I don't have any major complaints about this section. Only a short description of the research object is missing. This is related to cultural and infrastructural conditions. This is information that can contribute to assessing the possibility of using the method in other parts of the world.

Figure 1 is not very informative. Enlarge the photos and use a different layout – 2x2.

3. Results

The presentation of the results is correct. This is a consequence of the adopted methodology. I recommend technical fixes. Some graphs are not legible. Please enlarge them and use other layout.

4. Discussion

Correct discussion supported by examples of previous studies. An additional element could be recommendations for the design of correct architectural and technical solutions (in the context of the presented research).

5. Conclusions

The conclusions repeat the content of the previous sections. Conclusions should summarize the results and discussion, but not repeat them. It should be clearly justified that the manuscript contains sufficient contributions to the new body of knowledge from the international perspective. What new things (new theories, new methods or new policies) can the paper contribute to international literature? How to link the findings and conclusions in this paper with the previous findings and conclusions from other countries? All these questions must be clearly responded in Conclusion section.

References

The literature set is not impressive. Supplements should be made in accordance with the previous comments.

Conclusion from the review – the manuscript requires minor changes recommended by the reviewer.

Author Response

Thank you for your time and careful review.

We have had valuable comments from two reviewers and revised the manuscript based on the reviewers' comments. In some parts of the comments, one reviewer had different opinion from the other. So we have tried to revise the article from a neutral point of view to accept the constructive feedback of the reviewers. Here we have summarized the revisions and corrections. All changes are highlighted in red in revised article.

Once again, I appreciate your thorough review and look forward to receiving your response.

Reviewer 2 Report

Comments for Authors

Title :

Wrong title. You did not design vegetation. There is no design process of vegetation in the paper. The paper just created three green space concepts to test.

Therefore, your title seriously should be changed.      Perhaps you could paraphrase something like:

Effects of green space concepts…

Effect of forms of green roofs images….      Page one

              Abstract

The abstract discusses different things that the research. The research did not present any therapeutic landscape design activities.

There are different meanings in landscape design for “vegetation design types”.

The research just applied three abstract green space forms for the roof gardens.

Apparently, without correcting the professional terms and definitions, the research could mislead potential readers.                   

Please rewrite the abstract.             

Page 1

              Keywords:

there are wrong, unreverent, and misleading the researcher to find the paper. For example:

urban green: What does mean in your introduction? Do you mean urban green image? Perception? Percentage? Elements? Effects?

Therapeutic landscape: the research did not follow the therapeutic landscape design components and compounds.

Healing effects: it is hard to evaluate whether the research discovered any healing outputs. The research presented the relationships between the image of green spaces and the physiological reaction of the participants. Not any traces in the research about healing.

Green design: the research did not include any process of design. There is a conceptualization through simulation techniques in the research. However, there is no design in the research.

              Please change it with the proper keywords.        Page 1

1            Introduction:

1.1. The purpose of the research and motivation is clear in the introduction.

1.2 some information in the introduction did not develop: for example, 

“Van den Berg et al. (2014) compared the psychological effects of three different vegetation types by evaluating self-reported restoration levels [20]”.

Or

“Goto et al. (2013) measured changes in cardiac output in older adults when they were exposed to three different green space types [22]”.

What are those three? The introduction needs to clarify those components, otherwise, understanding your theoretical framework is difficult. 

While the introduction explained the purpose and gap the knowledge, the foundation of the research as a theoretical framework did not mention, which makes it difficult to understand the section.     

Please edit those missing parts of the information.           

Pages 2

2            Materials and methods

The methodology starts with the research design than methodology.

Therefore, the validity and reliability of the research design could not be confirmed in this way. It is highly recommended to improve the validity of the research design through an explanation of other research in the same alignment.

2.1 Visual stimuli

This section explains the research design. However, without a methodology section trust in the methods is so difficult.

The authors mentioned three types of green spaces were selected including grass, shrub, and tree. Nonetheless, the form, size and texture of the selected plants could seriously influence the perception of the users such as the contrast between columnar, pyramidal, and weeping trees that have different forms and effects. In addition, both size and scale of grasses are important. However, the image showed a grassed open field without details.

It is clear to me you used simulation and graphical techniques in the research design. However, the research is mute about it.

2.2 Subjects

Neither the research design nor methodology highlighted the reason why the researchers select men for the research than both genders.  

2.3 Data collection and analysis

This section represents the research process. However, for the first time, the reader realizes that the methodology of the research was designed based on the quantitative and statistical methods to apply ANOVA in SPSS software. Why ANOVA fit the data specification, did not reveal in the research. 

If you could please restructure the  section could be more understandable as following parts:

2.1: Methodology: related methods in similar research that other researchers applied and validity and reliability are clear.

2.2: Research design: the Visual Stimuli part seemingly is your research design. However, there are some questions that the research should answer in the methodology parts such as why the research did not reveal the simulation technique or quantitative method ANOVA.

I recommend reading some landscape architecture books and references to make your section more readable such as Architecture research methodologies by Linda Grout and David Wang, Landscape research design by Deming, and many other paper sources. 

2.3: Research process: what does the researcher take into action first, second, and so on (this part is mixed with the other parts and a major part of the Data collection and analysis is the research process)

2.4: Data Specification: (you mentioned you have done)

2.5: Statistical society and sampling methods (Despite the Subjects sections, the reason for the selection and specification of the participants did not discuss in the sections.)

2.6 implication and Limitation: if there is any error, missing, or limitation please explain in this part. (for example, the last paragraph of the discussion shows the research limitation)           Pages 2-3

3            Results

The results followed the ANOVA results presentation. It is informative.            

I recommend doing a reverse-engineering process and designing the methodology based on what you have done in the results.                Page 3-6

4            Discussion

3.1 The discussion did not link the theoretical framework (in the introduction) to challenge or approve the theories.

3.2 This section should be full of references to either support or challenge the results of the research. However, the section designed such as a research finding that strong contribution to the body of knowledge.

3.3 Discussion tried to make approval for the landscape design that the results could not show such kind influences and perhaps it create more confusion for the reader.

3.4 The application of the images to conclude a landscape theory is not reliable as the landscape design profession works with centimeters to design proper perceptions for the users.            

Please, do a back-forward between the deficiencies in the introductions and your results of the model to highlight a new bulk of knowledge that you are going to share.

Please challenge the previous theories or compare your findings with the previous outputs of the research to demonstrate how your results (and findings) could support the accumulation of knowledge or challenge them.

Please get help from a landscape architect who could help you to highlight the differentiation between composition-configuration, hardscape-softscape, and short and long views of the landscape and green spaces.            Page 6-7

5            Conclusion

The conclusion did not answer the introduction and what was the main problem of the research.             

Please revise the section and link the challenges and problems in the introduction with your analytical model of the analyses to answer the gap of knowledge.               Page 7

6            References

They are relevant.              There are more relevant studies have done recently on the effects of the landscape on pupils, and the therapeutic landscape in the psychological centers that could help you to develop landscape issues than a rough green image.               

7            Graphical figures

The quality of the images is poor.                 Page 3

Author Response

(The authors gave the same response as above.)
